# Fumigant Activity of Ethyl Formate against the Chestnut Weevil, *Curculio sikkimensis* Heller

**DOI:** 10.3390/insects13070630

**Published:** 2022-07-14

**Authors:** Tae Hyung Kwon, Byungho Lee, Junheon Kim

**Affiliations:** 1College of Agriculture and Life Science, Gyeongsang National University, Jinju 52828, Korea; xoxogudgud@naver.com; 2Institute of Agriculture and Life Science (BK21+ Program), Gyeongsang National University, Jinju 52828, Korea; byungholee@hotmail.com; 3Forest Entomology and Pathology Division, National Institute of Forest Science, Seoul 02455, Korea

**Keywords:** chestnut, *Castanea crenata*, fumigation, safety

## Abstract

**Simple Summary:**

Fumigation is the most effective method for the control of *Curculio sikkimensis* in chestnuts. The effects of ethyl formate (EF) as a fumigant were investigated to evaluate its potential for practical use by farmers. A dose of 180.0 g/m^3^ and 12 h of fumigation resulted in 100% mortality on a small scale (2 m^3^). The results of these experiments indicate that EF could be conveniently used as a fumigant by farmers.

**Abstract:**

*Castanea crenata* Siebold & Zucc. (Fagales: Fagaceae), a species of chestnut native to Korea and Japan, is distributed in Korea, Japan, and northeastern China, where chestnuts are a major economic agroforest product. *Curculio* spp. is among the main known pests of chestnuts around the world. In Korea, only phosphine (PH_3_) is permitted for the fumigation of *C. sikkimensis* larva-infested chestnuts. However, it is applied for large-scale fumigation, and its use is restricted. Moreover, it requires a long exposure time and an application device; thus, it cannot be used by small-scale farmers. In this study, the activity of ethyl formate (EF) as a fumigant against *Curculio sikkimensis* in chestnuts was investigated, and its potential for practical use by farmers was evaluated. The sorption of EF according to the filling ratio (FR) and fumigation time was tested, and the results revealed that 2.5% FR was the most effective. For *C. sikkimensis* in chestnuts, the mortality rate increased proportionately with the dose of EF. After exposure to 160 g/m^3^ of EF in a 12 L desiccator, the adult *C. sikkimensis* showed 100% mortality. According to the time–dose mortality data collected over 12 h of fumigation, the LCT_90_ and LCT_99_ values were estimated as 1052.0 and 1952.0 g·h/m^3^. The results revealed that immersion was not an effective method for controlling *C. sikkimensis*. According to the LCT values, a dose of 180.0 g/m^3^ and 12 h of fumigation resulted in 100% mortality on a small scale (2 m^3^). The results of this experiment indicate that EF could be conveniently used as a fumigant by farmers.

## 1. Introduction

*Castanea crenata* Siebold & Zucc. (Fagales: Fagaceae), a species of chestnut native to Korea and Japan, is distributed in Korea, Japan, and northeastern China [1,2]. Chestnuts are a major economic agroforest product, producing an annual income of ca. USD 88 billion [3]. *Curculio* spp. are among the main known pests of chestnuts around the world: *C. sikkimensis* in Korea, Japan, and China; *C. sayi* in the central–eastern US; and *C. propinquus* and *C. elephas* in Italy [4,5,6,7,8]. Unlike other seed pests, *Curculio* spp. (*C. sikkimensis*, *C. camelliae*, *C. robustus*) in Korea causes damage during storage after harvest and decreases the product value at the time of shipment, thus resulting in greater damage [9]. In order to reduce the damage caused by *C. sikkimensis* larva, different treatments have been studied, such as fumigation, immersion, and irradiation [10,11,12]. Although many studies have been performed, pesticides are favored and used by the growers or producers due to their cost and effectiveness. In Korea, only phosphine (PH_3_) is permitted for the fumigation of *C. sikkimensis* larva-infested chestnuts, as it was the only registered fumigant in Korea until 2020 [11]. However, it is used for large-scale fumigation, and its use is restricted. Moreover, it requires a long exposure time and an application device; thus, it cannot be used by small-scale farmers. Therefore, alternative fumigants must be evaluated and registered as pesticides for practical use by farmers.

In recent years, ethyl formate (EF) has been considered a candidate fumigant for combating insect pests of stored products. EF is a naturally occurring compound with relatively low toxicity toward mammals [13] and is recognized as a safe fumigant [14]. No residual problem has been identified, because it readily breaks down into naturally occurring products, ethanol and formic acid [15,16]. Vapormate^®^, which is the trade name of EF mixed with liquid CO_2_ in a proportion of 16.7 wt.%, has been registered in Australia since 22 March 2005 (Australian Pesticide and Veterinary Medicines Authority (APVMA) Reference #56186) [17] and in Korea for use against *Tetranychus urticae,* scale insects, aphids, and pests of stored products such as *Sitophilus zeamais* and *Lasioderma serricorne* [11]. In addition, liquid EF (active ingredient: 99%) is also registered in Korea for use against the same pests as Vapormate^®^. Experiments have been conducted using EF as a postharvest fumigant in the US, Australia, New Zealand, and Korea, showing good results [18,19,20,21,22].

This study aimed to investigate the activity of EF as a fumigant against *C. sikkimensis* in chestnuts and to evaluate its potential for practical use by farmers. In this study, *C. sikkimensis* larva-infested chestnuts were treated with EF for 12 h at 15 °C, and the concentration of sorption was measured to determine the appropriate filling ratio of fumigant for chestnuts. The mortality of *C. sikkimensis* resulting from a combination of EF and other treatments was also evaluated. Finally, the dose of EF for pesticide registration was determined.

## 2. Materials and Methods

### 2.1. Test Insects and Chestnuts

Chestnuts (*Castanea crenata* cv. Okkwang) infested by *C. sikkimensis* larvae were purchased from a local market, and were stored in a refrigerator at 10 °C until their use in experiments.

### 2.2. Chemicals

Ethyl formate (EF; 99% pure) was obtained from Safefume Pty Ltd., Hoengseong, Kangwon, Korea.

### 2.3. Quantitative Analysis Using Gas Chromatography

For the analysis of the EF concentration, 60 µL of the headspace fumigants from each test was subjected to a gas chromatograph (GC-17A, Shimadzu, Kyoto, Japan) equipped with a flame ionization detector (FID). The oven temperature was isothermal at 100 °C for 20 min, and the injector and detector temperatures were 250 and 280 °C, respectively. A nonpolar column, DB-5MS (30 m × 0.25 mm id., 0.25 µm film thickness; J&W Scientific. Folsom, CA, USA), was used. Helium was used as a carrier gas at a flow ratio of 1.5 mL/min. The peak area of the headspace content of EF in the desiccators compared to an external standard was used to determine the concentration in the desiccator’s headspace. The external standard curve was constructed by measuring 4 different concentrations (51.0–220.4 mg/L) in triplicate. 

### 2.4. Fumigation Bioassay for EF in Laboratory Conditions

A fumigation bioassay for EF was performed using *C. sikkimensis* larva-infested chestnuts and non-infested chestnuts at a 2.5% filling ratio in a desiccator (6.8 L) for 12 h at concentrations of 10.0–180.0 g/m^3^ at 15 °C. In total, 170 g of chestnuts was used, with half being infested by *C. sikkimensis* larvae (20 chestnuts/replicate). The desiccator (DWK Life Sciences, Mainz, Germany) was equipped with a septum injection system and sealed with glass stoppers containing filter paper. EF was applied on the filter paper through the septum using a micro-syringe. In the desiccator, *C. sikkimensis* lava-infested chestnuts and non-infested chestnuts were included. A magnetic bar was placed at the bottom of each desiccator to stir the fumigant for reducing the condensation of EF on the chamber surface. The experiments were performed in triplicate. The concentrations of the fumigants were monitored at multiple time intervals (10 min, 1 h, 2 h, 4 h, 8 h, and 12 h) and used to calculate the CT (concentration × time) values. The CT value according to the filling ratios was calculated according to the following equation described by Ren, et al. [23]:$$\mathrm{CT}=\sum \frac{\left(C_{i}+C_{i+1})(T_{i+1}-T_{i}\right)}{2}$$
where *C* is the concentration of the fumigant (g/m^3^), *T* is the time of exposure (h), *i* is the order of measurement, and CT is the concentration × time value.

The air in the desiccator was drawn using an airtight syringe, and the concentration of EF was analyzed using GC-FID (Shimadzu GC 17A, Shimadzu, Kyoto, Japan). The mortality of *C. sikkimensis* was checked by cutting the chestnut after fumigation. *C. sikkimensis* larvae were considered dead if any of the appendages did not move after air was blown onto them using an aspirator (Bug-Vac, Rose Entomology, AZO) and after a gentle push with fine-tipped forceps.

### 2.5. Sorption Test

For the identification of the sorption of EF on chestnuts, an experiment was conducted in laboratory conditions of 15 ± 1 °C and 40% RH. The desiccators (12 L) were filled with non-infested chestnuts (300–2400 g) to achieve different filling ratios (2.5%, 5.0%, 10.0%, and 20.0%) at 15 ± 1 °C and 80% RH. For the EF sorption test, a 160 g/m^3^ concentration of EF was applied to the filter paper inside the glass stopper and kept in the desiccator for 12 h under the above-described conditions. As a control, EF treatment was not performed. After 10 min, 1 h, 2 h, 4 h, 8 h, and 12 h, the air was drawn, and the concentration of EF was analyzed using GC-FID. The sorption rate was obtained as follows: sorption rate = (C_0_ − C*_n_*)/C_0_ (where C*_n_* is the concentration of EF at *n* h, and C_0_ is the initial concentration of EF).

For the phytotoxicity test, weight changes and the color value of the outer and inside of the chestnut, firmness, and decay degree were investigated at a CT value of 1064.7 g·h/m^3^ on the 7th day after fumigation with 10 chestnuts/replicates. These experiments were performed five times. Weight changes (%) were obtained using the following formula: 100 × (*Wo* − *Wn*)/*Wo*, where *Wo* is the weight of the chestnuts before fumigation ad *Wn* is the weight of the chestnuts on the 7th day after fumigation. The color value of the outer and inside of the chestnuts was monitored on the 7th day after fumigation using a color meter (ZE-2000, Nippon Denshoku Industries Co., Tokyo, Japan). Color values were obtained using the following formula: √L^2^ + a^2^ + b^2^ (L: degree of lightness; a: degree of redness; b: degree of yellowness). Firmness was measured using a fruit firmness tester (53,205 Digital fruit firmness tester, TR Turoni, Italy) equipped with an 8 mm steel plunger. Chestnuts were compressed to 6 mm in the equatorial zone at a rate of 0.5 mm s^−1^, and the maximum number developed during the operation was recorded. Results were expressed in kgf. The decay degree was classified as 0, 1, 2, 3, and 4 according to the proportion of decayed chestnut: 0: no decayed chestnuts; 1: <5% of chestnuts were decayed; 2: 5–25% of chestnuts were decayed; 3: 25–50% of chestnuts were decayed; 4: >50% of chestnuts were decayed.

### 2.6. Fumigation and Immersion Tests

To increase mortality, the fumigation and immersion methods were tested. An acrylic plate fumigation chamber (0.275 m^3^, 560 mm × 483 mm × 986 mm; fumigation chamber (As one, BG-1PT, Osaka, Japan)) was filled with *C. sikkimensis* lava-infested chestnuts (ca. 100 chestnuts/replicate) and non-infested chestnuts at a 2.5% filling ratio, and EF (120 g/m^3^; LCT_80_ value) was introduced into the chamber using a fumigator (Safefume Pty Ltd., Hoengseong, Korea) at 15 °C. The fumigator was used to fill liquid EF into the fumigation chamber by vaporizing it, and then the vaporized gas was injected into the fumigation chamber using nitrogen as the carrier gas. The concentrations of the fumigants were monitored at several time intervals (10 min, 1 h, 2 h, 4 h, 8 h, and 12 h) and used to calculate the CT (concentration × time) values. After fumigation, the fumigated chestnuts were immersed in water at 15 ± 1 °C for 12 h and dried at room temperature; then, the mortality of *C. sikkimensis* was checked (fumigation and immersion treatment). For the fumigation treatment, only EF was treated for 12 h under the same condition to conduct the fumigation and immersion treatment test without immersion into the water. The immersion treatment was performed by immersing the chestnuts (infested and non-infested) for 12 h without EF treatment. These experiments were performed in triplicate.

### 2.7. Depression (Reduced Pressure) Fumigation of Ethyl Formate

A depression fumigation chamber (55 L; vacuum desiccator, IKLab, Seoul, Korea) was filled with *C. sikkimensis* larva-infested chestnuts (ca. 100 chestnuts) and non-infested chestnuts at a 2.5% filling ratio, with reduced pressure at 7.99 kPa using a vacuum pump. EF, at a concentration of 120 g/m^3^ (LCT_80_ value), was used for treatment at 15 ± 1 °C. The concentrations of the fumigants were monitored at several time intervals (10 min, 1 h, 2 h, 4 h, 8 h, and 12 h) and used to calculate the CT (concentration × time) values. These experiments were performed in triplicate.

### 2.8. Small-Scale (2 m^3^) Fumigation Using Ethyl Formate

A hand-made LDPE film chamber (2 m^3^; 1 m (H) × 1 m (L) × 2 m (W)) equipped with a gas sampling line and a portable fan was filled with *C. sikkimensis* larva-infested chestnuts (ca. 300 chestnuts/replicate) and non-infested chestnuts at a 2.5% filling ratio. EF, at concentrations of 160 and 180 g/m^3^, was used for treatment for 12 h at 15 ± 1 °C. The concentrations of the fumigants were monitored at several time intervals (10 min, 1 h, 2 h, 4 h, 8 h, and 12 h) and used to calculate the CT (concentration × time) values. The air in the chamber was drawn into a Tedlar bag, and the concentration of EF was analyzed using GC-FID. The mortality of *C. sikkimensis* was checked by cutting the chestnuts on the 2nd day after fumigation. The experiments were performed in triplicate. According to the mortality, the control value was obtained as follows [24]:$$\mathrm{Control}\;\mathrm{value}\;\left(\%\right)=100\times \frac{\mathrm{Mortality}\;\%\;\mathrm{of}\;\mathrm{treatment}\;-\;\mathrm{Mortality}\;\%\;\mathrm{of}\;\mathrm{control}}{100-\mathrm{Mortality}\;\%\;\mathrm{of}\;\mathrm{control}}$$

### 2.9. Statistical Analysis

The mortality of each treatment was transformed to the arcsine square root and compared using one-way ANOVA. The means were compared and evaluated using the Tukey–Kramer honestly significant difference (HSD) test at the 0.05 significance level. The dose-dependent mortality data were subjected to Probit analysis to determine the lethal CT value of EF [25]. The means of sorption rate (SR) at each time interval were compared using one-way ANOVA and the Tukey–Kramer HST test at the 0.05 significance level. The means of each phytotoxicity data (weight change, color values, firmness, and decay degree) of the treatment and control were compared using the *t*-test at the 0.05 significance level. Statistical analysis was conducted using JMP ver. 9.02 (SAS Institute, Cary, NC, USA). Untransformed data are presented.

## 3. Results

### 3.1. Fumigation Bioassay Using EF in Laboratory Conditions

Mortality showed a dose-dependent response. Concentrations of EF above 110.0 g/m^3^ resulted in >90% mortality (Table 1). On the basis of the CT values and mortality, the lethal CT (LCT) value was obtained. When fumigation with EF was performed for 12 h, the LCT_50_, LCT_90_, and LCT_99_ values (confidential limit) were 530.3 (485.8–579.6), 1052.0 (848.6–1219.0), and 1952.0 (1629.0–2464.0) g·h/m^3^, respectively (slope ± SE = 3.4 ± 0.2, df = 14, χ^2^ = 20.4).

### 3.2. Sorption Test

The loss of EF in the headspace of a 6.8 L desiccator is shown in Figure 1, plotted as the ratio of headspace content to the injected dose (160 g/m^3^) against the fumigation time and expressed as the sorption ratio. When the filling ratios were increased, the sorption of EF also increased over time. Ten minutes after fumigation, 1.5%, 11.1%, 19.9%, and 28.7% of EF were absorbed, whereas 12 h after fumigation, 65.3%, 69.9%, 94.5%, and 98.9% of EF were absorbed for filling ratios of 2.5%, 5.0%, 10.0%, and 20.0%, respectively. The CT values for filling ratios of 2.5%, 5.0%, 10.0%, and 20.0% were 1067.7, 901.7, 369.5, and 236.6 g·h/m^3^, respectively. According to the results, a filling ratio of 2.5% is expected to be effective for controlling *C. sikkimensis*. Therefore, this filling ratio was implemented for subsequent experiments. There was no statistically significant difference between the EF treatment (CT value: 1064.7 g·h/m^3^) and the control in terms of weight change, color values of the outer and inside of the chestnut, firmness, and decay degree (Table 2).

### 3.3. Fumigation and Immersion Methods

The mortality following immersion for 12 h was low, at 1.6%. To evaluate their synergistic effect, immersion was performed after fumigation. Compared with the mortality following fumigation only, that following their combination did not show a statistically significant difference (Tukey’s HSD test; *p* = 0.05) (Table 3).

### 3.4. Depression (Reduced Pressure) Fumigation of Ethyl Formate

The mortality in a chamber that received only the reduced pressure of 7.99 kPa was 0%, whereas that in a chamber treated with EF for 12 h at the LCT_80_ value was 74.2%. EF treatment under reduced-pressure conditions (7.99 kPa) resulted in 73.2% mortality. There was a statistically significant difference between the control and both treatments (one-way ANOVA; *F*_3,8_ = 33.541, *p* < 0.001). However, the mortality following EF treatment at a normal pressure and reduced pressure did not exhibit a statistically significant difference (Tukey’s HSD test; *p* = 0.05).

### 3.5. Small-Scale (2 m^3^) Fumigation Using Ethyl Formate

To determine its potential for practical use, scale-up fumigation using EF was performed in a 2 m^3^ vinyl chamber. When chestnuts infested with *C. sikkimensis* lava were fumigated using EF at 160.0 and 180.0 g/m^3^, the mortality of *C. sikkimensis* was 91.42% and 100%, respectively (Table 4). The control value of EF at 160.0 and 180.0 g/m^3^ was 91.4% and 100%, respectively (Table 4).

## 4. Discussion

Our results demonstrate that EF fumigation for 12 h at 15 °C could effectively control (>90% mortality) *C. sikkimensis* larvae in chestnuts without any apparent negative impact on their quality at concentrations of 110.0 and 130.0 g/m^3^. In a small-scale trial using a 2 m^3^ LDPE fumigation chamber, a scheduled dose of 180 g/m^3^ at 15 °C for 12 h with a 2.5% chestnut filling ratio resulted in the complete control of *C. sikkimensis* larvae inside the chestnut. Thus, a dose of 180 g/m^3^ was determined for pesticide registration.

It is well known that the concentration of fumigant was affected by different life stages of a target insects [26,27]. In the case of *C. sikkimensis*, the adult lays eggs on chestnuts from summer to autumn, the larva grows up by consuming the chestnuts and the late larva escapes from the chestnut for pupation [12]. Therefore, the larva exists in the chestnut during harvest time and damages the chestnut. In these experiments, the larva of *C. sikkimensis* was treated with EF, which is a major concern to be addressed.

For controlling coleopteran pests of stored products, Agarwal, et al. [28] reported that 24 h of fumigation at 22–24 °C using EF at 25–30 g/m^3^ without apples could achieve the complete control of eucalyptus weevils, *Gonipterus platensis* (Coleoptera: Curculionidae), while EF at 40 g/m^3^ under the same conditions, with a 90–95% filling ratio of apples, could achieve complete control. Kim, et al. [29] reported that the LCT_99_ values for *Sitophilus oryzae* (Coleoptera: Curculionidae) exposed to EF at 20 °C were 775.48 and 449.20 g·h/m^3^ for phosphine-susceptible and -resistant late larvae, respectively. Asimah, et al. [30] reported that fumigation using 190 g/L of liquid EF (CT value = 1322.12 g·h/L) for 40 h was sufficient to control all the stages of *Tribolium castaneum* (Coleoptera: Tenebrionidae) and *Lasioderma serricorne* (Coleoptera: Anobiiadae) in cocoa beans. In the case of other insect pests, which live on the surface of fruits, the LCT values were lower than those for coleopteran pests of stored products. For mealybugs, *Planococcus citri* (Hemiptera: Pseudococcidae), the LCT_90_ and LCT_99.9_ values of EF were reported to be 46.25 and 72.15 g/m^3^ following 2 h of fumigation at 13 °C [31]. For *Tetranychus urticae*, 4 h of fumigation using 60 g/m^3^ of EF resulted in 100% control [32]. The differences in susceptibility among insects may result from differences in terms of habitat, i.e., inside fruits vs. on their surface.

The EF sorption in chestnuts increased with the filling ratio from 2 to 20%. This phenomenon has also been reported in other products, such as sweet pumpkins [32] and blueberries [26]. However, the sorption ratio in chestnuts was much lower than that in sweet pumpkins at ca. 0.5 following 4 h of exposure at a filling ratio of 21% [32]. EF is soluble in water, and it seems to be highly adsorbed on chestnuts in this medium. Harvested chestnuts are usually preserved in a lower-temperature chamber (ca. 3–5 °C), while the fumigation treatment was performed at ca. 15 °C. In this experiment, chestnuts stored in a refrigerator at 10 °C were used for the fumigation treatment at ca. 15 °C. This temperature difference led to condensation on the chestnut surface. Thus, increasing the filling ratio would have increased the level of dew on the surface.

To increase the efficacy of fumigation, combined methods, including fumigation + immersion and depression fumigation, were performed. Unfortunately, none of these methods increased the efficacy of fumigation. Kim et al. [12] reported that immersion treatment for 10–24 h at <20 °C did not yield a satisfactory effect of control (mortality: 0–28.8%), whereas treatment for >10 h at >30 °C resulted in the complete control of *C. sikkimensis*. However, they also reported that a longer immersion time and higher temperature affected the quality of the chestnuts; for example, they resulted in a higher decay ratio, lower hardness, higher moisture content, and lower free-sugar content. Thus, we performed fumigation and immersion in water at 15 ± 1 °C. Under vacuum conditions, no mortality of *C. sikkimensis* was observed. Consequently, the mortality of fumigated *C. sikkimensis* was due to the presence of EF. Following reduced-pressure fumigation, there was no statistically significant difference in mortality between vacuum conditions and normal conditions. However, Misumi, et al. [33] reported that the mortality of *Planococcus citri* eggs was increased under vacuum conditions by up to 61.2–100% upon increasing the dosage ratio. Stewart, et al. [34] reported that a smaller amount (0.5%) of EF had the same control effect as a larger amount of EF when applied for treatment under vacuum conditions. Some other combination treatments, such as fumigation + cold treatments and combined fumigants, have also been studied to increase the efficacy of fumigation. Kwon et al. [26] found that treatments with low-dose EF (LCT 50% level) prior to cold treatment increased the efficacy of control of pupae, which was not sufficiently controlled by only cold treatment. A better control result was achieved with a combined treatment of EF and phosphine (PH_3_) against *Lasioderma serricorne* (Coleoptera: Anobiidae) at lower concentrations than for EF treatment alone and at lower exposure times than for phosphine treatment alone [22]. Although combined fumigants were not tested in this study, the combination of EF and PH_3_ is worth investigating.

In our experiments, the fumigation time and temperature were fixed at 12 h and 15 °C, respectively, because the farmers or producers usually fumigate for 12 h for convenience, and this is typical temperature at chestnut harvest time. Thus, EF fumigation for 12 h at 15 °C would be practical in Korea. However, for the application of EF fumigation against another *Curculio* spp. or in another place, a different fumigation time and temperature would need to be studied.

## 5. Conclusions

On the basis of the results of this study, fumigation with ethyl formate (EF) at a dose of 180.0 g/m^3^ for 12 h resulted in 100% mortality against *C. sikkimensis* larvae in chestnuts on a small scale (2 m^3^). The results of this experiment indicate that EF could be a promising fumigant with which to control *C. sikkimensis* in chestnuts and could be conveniently used as a fumigant by farmers.

## Figures and Tables

**Figure 1 insects-13-00630-f001:**
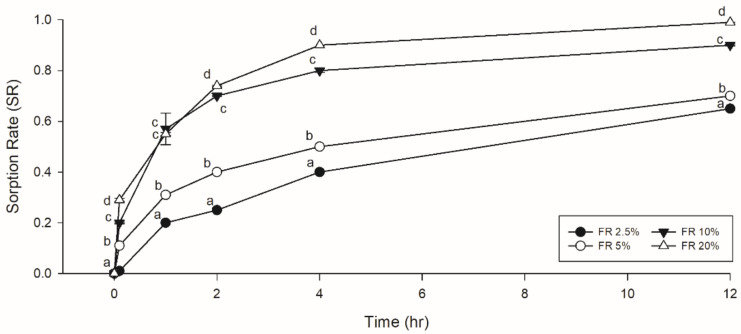
Sorption of ethyl formate according to filling ratio (FR) and fumigation time, expressed as the sorption rate (SR; mean ± SE), which is the ratio of the headspace content of EF at a specified time of fumigation (*C*_0_–*C_n_*) to the initial applied concentration (*C*_0_). Means at each timepoint followed by the same letters are not significantly different (Tukey’s HSD test; *p* = 0.05).

**Table 1 insects-13-00630-t001:** Mortality of *Curculio sikkimensis* and CT values after 12 h of fumigation at 15 °C (filling ratio = 2.5%).

Concentration of EF (g/m^3^)	Number of Larvae	Mortality (Mean ± SE, %)	CT Value of EF (g·h/m^3^)
0.0	60	0.0 ± 0.0	0.0
10.0	60	0.0 ± 0.0	64.5
20.0	60	13.3 ± 1.7	151.6
30.0	60	21.7 ± 1.7	248.6
40.0	60	41.7 ± 1.7	359.5
60.0	60	55.0 ± 2.9	597.5
80.0	60	76.7 ± 3.3	729.9
90.0	60	86.7 ± 1.7	820.4
110.0	60	91.7 ± 1.7	964.3
130.0	60	96.7 ± 1.7	1120.8
160.0	60	100.0 ± 0.0	1221.9
180.0	60	100.0 ± 0.0	1360.3

**Table 2 insects-13-00630-t002:** Phytotoxic effects of fumigation on chestnut for 12 h at 15 °C (filling ratio = 2.5%).

CT Value (g·h/m^3^)	Weight Change (%)	Color Values ^1^		Firmness (kgf)	Decay Degree ^2^
		Outer	Inside		
0	1.7 ± 0.8 ^a^	46.3 ± 1.6 ^a^	23.5 ± 1.3 ^a^	6.5 ± 1.1 ^a^	0.0 ± 0.0 ^a^
1064.7	1.5 ± 0.5 ^a^	49.3 ± 0.7 ^a^	24.6 ± 1.1 ^a^	6. 4 ± 0.6 ^a^	0.0 ± 0.0 ^a^

^1^ Color values: √L^2^ + a^2^ + b^2^ (L: degree of lightness; ^a^: degree of redness; ^b^: degree of yellowness). ^2^ Decay degree: 0: no spoiled chestnut; 1: <5% of chestnuts were decayed; 2: 5–25% of chestnuts were decayed; 3: 25–50% of chestnuts were decayed; 4: >50% of chestnuts were decayed. Means within a column followed by the same letters are not significantly different (*t*-test; *p* = 0.05).

**Table 3 insects-13-00630-t003:** Effects of fumigation and immersion on *Curculio sikkimensis* at 15 °C (filling ratio = 2.5%).

Treatment	Time (h)	CT (g·h/m^3^)	Mortality (Mean ± SE, %)	Numbers of Larvae
Immersion	12	-	1.6 ± 0.3 ^b^	309
Fumigation ^1^	12	838.6	78.8 ± 4.2 ^a^	302
Fumigation ^1^ and immersion	12/12	838.6	76.9 ± 8.2 ^a^	315
Control	-	-	0.0 ^b^	301

^1^ Fumigation: ethyl formate at 120 g/m^3^ (LCT_80_ value). Means within a column followed by the same letters are not significantly different (Tukey’s HSD test; *p* = 0.05).

**Table 4 insects-13-00630-t004:** Mortality of *Curculio sikkimensis* (ca. 300 larvae/replicate, N = 3) after 12 h of fumigation using ethyl formate (EF) at 15 °C (filling ratio = 2.5%) on a small scale (2 m^3^).

**Treatment**	**Number of Larvae**	**Mortality (** **Mean ± SE, %)**	**Control Value (%)**
EF 160.0 g/m^3^	963	91.42 ± 1.05	91.4
EF 180.0 g/m	900	100.0	100.0
Control	911	0.11 ± 0.11	-

## Data Availability

Data is contained within the article.

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
