# Peer review of "Fumigant Activity of Ethyl Formate against the Chestnut Weevil, Curculio sikkimensis Heller"

_insects, 2022, doi:10.3390/insects13070630_

Round 1
Reviewer 1 Report
The manuscript isgenerally fine for me, please find some necessary corrections below:
lines 19, 39: the correct name is Castanea sikkimensis
lines 44-46: This is not true in general: For instance, Curculio nutum damage seeds in late summer and as such, it not an storage pest. Must be corrected.
line 48: „pesticides are favored and used by customers”: pesticides are used by the growers or producers (somewhere in themarkating chain) but notby customers
line 118: What does phytotoxicity mean? How was it tested? In 3.2 (l. 178 – 180) „outer and inside color of the chestnut, hardness, and spoilage degree” is mentioned, but data are not presented. These latter are not phytotoxicity tests. If these findings are mentioned then must be supported by data or by references from other sources.
line 118: „at 7 days after…” correct: at 7th day after fumigation
Table 2 is broken on two pages (4 – 5), it must be re-edited
3.3: What was the theory behind the immersion the chestnuts in the water? As a general rule, such products must be stored dry. Even a much shorter contact with water impose the risk of infection by moulds and production of mycotoxins as a consequence.
Author Response
Responses to Reviewers’ comments
We appreciate the reviewers spending time and efforts to review our manuscript and thank for the valuable comments and suggestions that could improve our manuscript.
We hope that the concerns have been resolved with the modifications and explanations given.
Reviewer 1 comments
The manuscript is generally fine for me, please find some necessary corrections below:
lines 19, 39: the correct name is Castanea sikkimensis
[Response] Scientific name of chestnut is Castanea crenata thus it was corrected as Castanea crenata through the manuscript.
lines 44-46: This is not true in general: For instance, Curculio nutum damage seeds in late summer and as such, it not an storage pest. Must be corrected.
[Response] All Curculio spp. in Reference 9 were listed and damage was limited to Korea.
line 48: „pesticides are favored and used by customers”: pesticides are used by the growers or producers (somewhere in the markating chain) but not by customers
[Response] By the growers or producers were correct. It changed as by the growers or producers
line 118: What does phytotoxicity mean? How was it tested? In 3.2 (l. 178 – 180) „outer and inside color of the chestnut, hardness, and spoilage degree” is mentioned, but data are not presented. These latter are not phytotoxicity tests. If these findings are mentioned then must be supported by data or by references from other sources.
[Response] We described details of phytotoxicity test in M&M section. The results were described in Table 3.
line 118: „at 7 days after…” correct: at 7th day after fumigation
[Response] It changed as commented and also 2 days after fumigation was changed as 2nd day after fumigation.
Table 2 is broken on two pages (4 – 5), it must be re-edited
[Response] It was re-edited
3.3: What was the theory behind the immersion the chestnuts in the water? As a general rule, such products must be stored dry. Even a much shorter contact with water impose the risk of infection by moulds and production of mycotoxins as a consequence
[Response] The immersion cause mortality of the insect was reported by Kim et al. (Ref. 9). Conventionally, the farmers or producers immersed the chestnut to select C. sikkimensis-infected chestnut. Because immersion should be performed, we wondered the immersion and fumigation could increase the efficacy of fumigation. As you indicated, the chestnuts were decayed much, and lowered the hardness.
Reviewer 2 Report
Manuscript ID: insects-1777205
Title: Fumigant Activity of Ethyl Formate against the Chestnut Weevil, Curculio sikkimensis Heller
Authors: Tae Hyung Kwon, Byung-Ho Lee, Junheon Kim
General comments:
This manuscript reported a study on efficacy of ethyl formate against chestnut weevil. The study is reasonably well designed and conducted and an effective treatment was determined. Most results were also properly analyzed, presented, and discussed. Although parts of experiments on immersion and vacuum fumigation did not increased efficacy of EF fumigation, results can still be beneficial to other researchers. The outcome of the study has practical implication for management of the pest by farmers. The manuscript in general is well written. However, the manuscript still has several major concerns and a major revision is needed to address these concerns before it can be published. Below is a list of major concerns need to be addressed.
For sorption test, phytotoxicity was examined but without any details of the method used and what quality parameters were examined. Please add some details here corresponding to results.
It is misleading to define sorption rate as Cn/C0 (EF concentration at n h over initial EF concentration) as described in Sorption test and presented in Fig 1 since higher values actually mean lower amounts of EF adsorbed in chestnuts. More accurately it should be called Relative reduction of EF concentration. I suggest to redefine sorption rate as reduction of EF concentration over the initial EF concentration [(C0-Cn)/C0]. Sorption rate can also be presented as amount of EF adsorbed per unit weight of chestnuts (g/kg) over time.
There is a lack of rationale to best combinations of EF fumigation and immersion to increase efficacy. Is there evidence that 12 h immersion can cause mortality of the insect (cite references if there are)? However, since EF is highly soluble, would immersion help to desorb EF from fumigated chestnuts and therefore beneficial? Or is it necessary to desorb EF from chestnuts?
Small-scale fumigation tests: Please provide the total number of insects tested.
Tables 1, 3, 4: Please provide numbers of insects tested for each treatment.
Delete Table 2 and describe results in text.
Author Response
Responses to Reviewers’ comments
We appreciate the reviewers spending time and efforts to review our manuscript and thank for the valuable comments and suggestions that could improve our manuscript.
We hope that the concerns have been resolved with the modifications and explanations given.
Reviewer 2 comments
General comments:
This manuscript reported a study on efficacy of ethyl formate against chestnut weevil. The study is reasonably well designed and conducted and an effective treatment was determined. Most results were also properly analyzed, presented, and discussed. Although parts of experiments on immersion and vacuum fumigation did not increased efficacy of EF fumigation, results can still be beneficial to other researchers. The outcome of the study has practical implication for management of the pest by farmers. The manuscript in general is well written. However, the manuscript still has several major concerns and a major revision is needed to address these concerns before it can be published. Below is a list of major concerns need to be addressed.
For sorption test, phytotoxicity was examined but without any details of the method used and what quality parameters were examined. Please add some details here corresponding to results.
[Response] We described details of phytotoxicity test in M&M section. The results were described in Table 3.
It is misleading to define sorption rate as Cn/C0 (EF concentration at n h over initial EF concentration) as described in Sorption test and presented in Fig 1 since higher values actually mean lower amounts of EF adsorbed in chestnuts. More accurately it should be called Relative reduction of EF concentration. I suggest to redefine sorption rate as reduction of EF concentration over the initial EF concentration [(C0-Cn)/C0]. Sorption rate can also be presented as amount of EF adsorbed per unit weight of chestnuts (g/kg) over time.
[Response] As the reviewer suggested, sorption rate was changed
There is a lack of rationale to best combinations of EF fumigation and immersion to increase efficacy. Is there evidence that 12 h immersion can cause mortality of the insect (cite references if there are)? However, since EF is highly soluble, would immersion help to desorb EF from fumigated chestnuts and therefore beneficial? Or is it necessary to desorb EF from chestnuts?
[Response] The immersion cause mortality of the insect was reported by Kim et al. (Ref. 9). Conventionally, the farmers or producers immersed the chestnut to select C. sikkimensis-infected chestnut. Because immersion should be performed, we wondered the immersion and fumigation could increase the efficacy of fumigation.
Small-scale fumigation tests: Please provide the total number of insects tested.
[Response] The total number of insects tested was described as C. sikkimensis-infected chestnuts (300 chestnuts) with 3 replicates. Thus 900 C. sikkimensis larvae (= chestnut) was tested. In Table 4. The total numbers of tested was described.
Tables 1, 3, 4: Please provide numbers of insects tested for each treatment.
[Response] Numbers of insect tested was described
Delete Table 2 and describe results in text.
[Response] Done as suggested
Round 2
Reviewer 2 Report
Additional minor changes are needed.
More details are needed in Statistical Analysis section for analyses of data on phytotoxicity, sorption, and combinations of EF fumigation and immersion. Right now, only analysis of insect mortality data is included.
Sorption rate in percentage should be calculated as (C0-Cn)/C0x100. Please revise it in method and Fig 1 caption. The reduction of EF concentration by time n is C0-Cn not Cn-C0.
Table 3, move the column of number of larvae to next to mortality column.
Table 4, combine Treatment and Concentration into one column. Also remove the control value column or explain it in the results.
Author Response
Responses to Reviewers’ comments
We appreciate the reviewer spending time and efforts to review our manuscript and thank for the valuable comments and suggestions that could improve our manuscript.
We hope that the concerns have been resolved with the modifications and explanations given.
More details are needed in Statistical Analysis section for analyses of data on phytotoxicity, sorption, and combinations of EF fumigation and immersion. Right now, only analysis of insect mortality data is included.
[Response] We added the analysis methods of sorption rate and phytotoxicity.
The data of combination of EF fumigation and immersion was mortality; therefore, it was not described.
Sorption rate in percentage should be calculated as (C0-Cn)/C0x100. Please revise it in method and Fig 1 caption. The reduction of EF concentration by time n is C0-Cn not Cn-C0.
[Response] (Cn-Co)/Co was type-missed. It was calculated as (Co-Cn)/Co, and
it was corrected in the revised manuscript (Text and Figure 1 caption).
Table 3, move the column of number of larvae to next to mortality column.
[Response] Changed as suggestion
Table 4, combine Treatment and Concentration into one column. Also remove the control value column or explain it in the results.
[Response] Treatment and concentration were combined.
The control value was explained in M&M section (line 163-165) and also described in the results section.